# Isospectral Twirling and Quantum Chaos

**DOI:** 10.3390/e23081073

**Published:** 2021-08-19

**Authors:** Lorenzo Leone, Salvatore F. E. Oliviero, Alioscia Hamma

**Affiliations:** Physics Department, University of Massachusetts Boston, Boston, MA 02125, USA; S.Oliviero001@umb.edu (S.F.E.O.); alioscia.hamma@umb.edu (A.H.)

**Keywords:** quantum chaos, information scrambling, entanglement, twirling

## Abstract

We show that the most important measures of quantum chaos, such as frame potentials, scrambling, Loschmidt echo and out-of-time-order correlators (OTOCs), can be described by the unified framework of the isospectral twirling, namely the Haar average of a *k*-fold unitary channel. We show that such measures can then always be cast in the form of an expectation value of the isospectral twirling. In literature, quantum chaos is investigated sometimes through the spectrum and some other times through the eigenvectors of the Hamiltonian generating the dynamics. We show that thanks to this technique, we can interpolate smoothly between integrable Hamiltonians and quantum chaotic Hamiltonians. The isospectral twirling of Hamiltonians with eigenvector stabilizer states does not possess chaotic features, unlike those Hamiltonians whose eigenvectors are taken from the Haar measure. As an example, OTOCs obtained with Clifford resources decay to higher values compared with universal resources. By doping Hamiltonians with non-Clifford resources, we show a crossover in the OTOC behavior between a class of integrable models and quantum chaos. Moreover, exploiting random matrix theory, we show that these measures of quantum chaos clearly distinguish the finite time behavior of probes to quantum chaos corresponding to chaotic spectra given by the Gaussian Unitary Ensemble (GUE) from the integrable spectra given by Poisson distribution and the Gaussian Diagonal Ensemble (GDE).

## 1. Introduction

The onset of chaotic dynamics is at the center of many important phenomena in quantum many-body systems, from thermalization in a closed system [1,2,3,4,5,6,7,8,9,10,11] to scrambling of information in quantum channels [12,13,14,15,16,17], black hole dynamics [18,19,20,21], entanglement complexity [22,23], to pseudorandomness in quantum circuits [24,25,26,27,28], and finally, the complexity of quantum evolutions [29,30,31,32]. Several probes of quantum chaos have been studied in recent years [33,34,35,36]. Chaos, equilibration, thermalization and other related phenomena are described by the behavior of entanglement growth and typicality, the Loschmidt echo, and out-of-time-order correlation functions (OTOCs) [20,37,38,39,40,41,42,43,44,45]. Information scrambling is characterized by tripartite mutual information [12,13] and its connection OTOCs. Pseudorandomness is characterized by the frame potential, which describes the adherence to moments of the Haar measure [46,47]; the complexity of entanglement is characterized by the adherence to the random matrix theory distribution of the gaps in the entanglement spectrum [22,23].

Random matrix theory (RMT) has been extensively studied and applied to quantum chaos [48,49,50,51,52,53]. The quantization of classical chaotic systems has often resulted in quantum Hamiltonians with the same level of spacing statistics of a random matrix taken from the Gaussian Unitary Ensemble (GUE). One could take the behavior of OTOCs, entanglement, frame potentials and other probes under a time evolution induced by a chaotic Hamiltonian, e.g., a random Hamiltonian from GUE, and define it as the characteristic behavior of these quantities for quantum chaos [54,55,56]. Though we agree with the heuristics of this approach, it would be important to compare the time behavior of these probes in systems that are not characterized by a spectrum given by a random matrix, or on the other hand, by Hamiltonians whose eigenvectors are not a random basis according to the Haar measure, e.g., Hamiltonians with eigenvectors that, although possessing high entanglement, do not contain any magic, that is, they are stabilizer states. Attempts at showing the difference in behavior between chaotic and non-chaotic behavior are often limited to specific examples [38,57]. Moreover, given the proliferation of probes to quantum chaos, one does feel the necessity of having a unified framework to gather together all these results.

In this paper, we set out to provide such a unifying framework and to prove that one can clearly distinguish chaotic from non-chaotic dynamics. The framework is provided by the *isospectral twirling* ${\hat{R}}^{\left(2k\right)}\left(U\right)$, that is, the Haar average of a k−fold channel. This operation randomizes over the eigenstates of a unitary channel *U* but leaves the spectrum invariant. In this way, one obtains quantities that are functions of the spectrum only. The unitary channel represents the quantum evolution induced by a Hamiltonian. Chaotic Hamiltonians feature spectra obeying the random matrix theory, e.g., GUE, while integrable systems possess spectra obeying other statistics [58,59,60,61]. The main results of this paper are: (i) the isospectral twirling unifies all the fundamental probes P used to describe quantum chaos in the form of ${\langle P_{O}\rangle }_{G}=\mathrm{tr}\left[\tilde{T}O{\hat{R}}^{\left(2k\right)}\right]$, where $\tilde{T}$ is a rescaled permutation operator, O characterizes the probe, ${\langle \cdot \rangle }_{G}$ is the Haar average and (ii) by considering the isospectral twirling associated to a k−doped Clifford group C(d), we show that the asymptotic temporal behavior of the OTOCs interpolates between a class of integrable models and quantum chaos and does not depend on the specific spectrum of the Hamiltonian; (iii) finally, by computing the isospectral twirling for the spectra corresponding to the chaotic Hamiltonians in GUE and integrable ones—Poisson, Gaussian Diagonal Ensemble (GDE)—the isospectral twirling can distinguish chaotic from non-chaotic behavior in the temporal profile of the probes, though all the spectra lead to the same asymptotic behavior—a sign of the fact that chaos is not solely determined by the spectrum of the Hamiltonian but also by its eigenvectors.

## 2. Isospectral Twirling

Let $H≃C^{d}$ be a d−dimensional Hilbert space and let U∈U(H) with spectral resolution $U=\sum_{k}e^{-iE_{k}t}\Pi_{k}$, where $\Pi_{k}$ are orthogonal projectors on H. We can think of the $\mathrm{Sp}\left(H\right)\equiv {\left\{E_{k}\right\}}_{k=1}^{d}$ as the spectrum of a Hamiltonian *H*. Through *H* we can generate an isospectral family of unitaries $E_{H}\equiv {\left\{U_{G}\left(H\right)\right\}}_{G}:=\left\{G^{†}\exp\left\{-iHt\right\}G,\;G\in U\left(H\right)\right\}$. Define *isospectral twirling* as the 2k−fold Haar channel of the operator $U^{⊗k,k}\equiv U^{⊗k}⊗U^{†⊗k}$ by
(1)$$\begin{array}{c} {\hat{R}}^{\left(2k\right)}\left(U\right):=\int \;dG\;G^{†⊗2k}\left(U^{⊗k,k}\right)G^{⊗2k} \end{array}$$
where dG represents the Haar measure over U(H). This object has been previously used to demonstrate convergence to equilibrium under a random Hamiltonian [62] or the behavior of random quantum batteries [63]. Under the action of (Equation 1), the spectrum of *U* is preserved. A general way to compute the above average is to use the Weingarten functions [64]. We obtain:(2)$$\begin{array}{c} {\hat{R}}^{\left(2k\right)}\left(U\right)=\sum_{\pi \sigma }{\left({\tilde{\Omega }}^{-1}\right)}_{\pi \sigma }\mathrm{tr}\left({\tilde{T}}_{\pi }^{\left(2k\right)}U^{⊗k,k}\right){\tilde{T}}_{\sigma }^{\left(2k\right)} \end{array}$$
where ${\tilde{T}}_{\pi }^{\left(2k\right)}\equiv T_{\pi }^{\left(2k\right)}/d_{\pi }^{\left(2k\right)}$, $\pi ,\sigma \in S_{2k}$ are (rescaled) permutation operators of order 2k, $d_{\pi }^{\left(2k\right)}=\mathrm{tr}T_{\pi }^{\left(2k\right)}$ and ${\left({\tilde{\Omega }}^{-1}\right)}_{\pi \sigma }\equiv {\left[\mathrm{tr}\left({\tilde{T}}_{\pi }^{\left(2k\right)}{\tilde{T}}_{\sigma }^{\left(2k\right)}\right)\right]}^{-1}$ are rescaled Weingarten functions. Notice that, through *U*, the isospectral twirling is a function of the time *t*.

## 3. The Integrability-Chaos Transition

The Hamiltonian generating the unitary temporal evolution in a closed quantum system can be written in its spectral resolution $H=\sum_{i}E_{i}\Pi_{i}$, showing explicitly that the dynamics are contained both in the eigenvalues and the eigenvectors of *H*. We now show that information about the asymptotic temporal behavior is contained in the way we pick the eigenvectors of *H*. To this end, consider a system of *N* qubits, $H=C^{2^{N}}$, and a Hamiltonian diagonal in the computational basis ${\left\{|i\rangle \right\}}_{i=1}^{2^{N}}$, namely $H=\sum_{i}E_{i}\Pi_{i}$ with $\Pi_{i}=|i\rangle \langle i|$ orthogonal projectors. Define then the average asymptotic unitary
(3)$$U_{\infty }^{⊗2,2}:=\lim_{t\to \infty }{\bar{U^{⊗2,2}}}^{P(E_{i})}$$
where the average is taken over a Schwartzian probability distribution of spectra. The isospectral twirling of $U_{\infty }^{⊗2,2}$ thus does not depend on the distribution of the eigenvalues $P(E_{i})$.

We now map these projectors by $\Pi_{i}\mapsto C\Pi_{i}C^{†}$ with C∈C(d) the Clifford group. These projectors are not typical in the Hilbert space and they cannot be clearly associated with chaotic behavior, for instance, it is not clear whether they feature ETH. They would possess typical entanglement, but its fluctuations obtained by the Haar measure on the unitary group are not the same. Define the Cl−Isospectral twirling for $U_{\infty }^{⊗2,2}$ as
(4)$$R_{Cl}^{\left(4\right)}\left(U_{\infty }\right):=\int_{C(2^{N})}dCC^{†⊗4}U_{\infty }^{⊗2,2}C^{⊗4}$$A general way to compute the Clifford average of order four is to use the generalized Weingarten functions formula, which is a rearrangement of the formula shown in [65]:(5)$$\begin{array}{ccc} \;R_{Cl}^{\left(4\right)}\left(U_{\infty }\right)\; & = & \;\sum_{\pi \sigma }W_{g}^{+}\left(\pi \sigma \right)\mathrm{tr}\left(U_{\infty }^{⊗2,2}QT_{\sigma }\right)QT_{\pi }\;\\ \; & + & \;W_{g}^{-}\left(\pi \sigma \right)\mathrm{tr}\left(U_{\infty }^{⊗2,2}Q^{⊥}T_{\sigma }\right)Q^{⊥}T_{\pi } \end{array}$$
where $Q=\frac{1}{d^{2}}\sum_{P\in P(2^{N})}P^{⊗4}$, $Q^{⊥}={1\;l}^{⊗4}-Q$ and $P\in P(2^{N})$ elements of the Pauli group on *N*-qubits, while
(6)$$W_{g}^{\pm }\left(\pi \sigma \right):=\sum_{\lambda \;|\;D_{\lambda }^{\pm }\neq 0}\frac{d_{\lambda }^{2}}{{\left(4!\right)}^{2}}\frac{\chi^{\lambda }\left(\pi \sigma \right)}{D_{\lambda }^{\pm }}$$
here λ labels the irreducible representations of the symmetric group $S_{4}$, $\chi^{\lambda }\left(\pi \sigma \right)$ are the characters of $S_{4}$ depending on the irreducible representations λ, $d_{\lambda }$ is the dimension of the irreducible representations λ, $D_{\lambda }^{+}=\mathrm{tr}\left(QP_{\lambda }\right)$ and $D_{\lambda }^{-}=\mathrm{tr}\left(Q^{⊥}P_{\lambda }\right)$ where $P_{\lambda }$ are the projectors onto the irreducible representations of $S_{4}$, and finally, $T_{\sigma }$ are permutation operators corresponding to the permutation $\sigma \in S_{4}$.

Let the unitary evolution be generated by a l−doped Hamiltonian $H_{l}=C^{(l)†}H_{0}C^{\left(l\right)}$ where
(7)$$\begin{array}{c} C^{\left(l\right)}=\prod_{r}C_{r}^{†}K_{r} \end{array}$$In the equation above, every $C_{r}\in C\left(d\right)$ is an element of the Clifford group, while $K_{r}$ is a single qubit gate not belonging to the Clifford group. In this way, we have doped the Clifford Hamiltonian with non-Clifford resources. Notice that, for l=0, the Hamiltonian is the sum of commuting Pauli strings, and it is therefore integrable. We want to show that by inserting the gates $K_{r}$, we obtain a transition to quantum chaotic behavior. This result would also show that integrability can be deformed in a “smooth” way and attain a crossover to quantum chaos. To this end, we use a particular probe to quantum chaos given by the OTOCs, as we shall see in the next section.

## 4. OTOCs

Scrambling of information can be measured by two quantities, the OTOCs [33] and the tripartite mutual information (TMI) [13]; namely, the decay of the OTOC implies the decay of the TMI [12,13]. In this section, we show how the OTOCs are described by the isospectral twirling. Consider 2k local, non-overlapping operators $A_{r}$, $B_{r}$, r∈[1,k]. The infinite temperature 4k−point OTOC is defined as
(8)$$\begin{array}{ccc} {\mathrm{OTOC}}_{4k}\left(t\right) & = & d^{-1}\mathrm{tr}(A_{1}^{†}\left(t\right)B_{1}^{†}\cdots A_{k}^{†}\left(t\right)B_{k}^{†}\\ & \times & A_{1}\left(t\right)B_{1}\cdots A_{k}\left(t\right)B_{k}) \end{array}$$
where $A_{l}\left(t\right)=e^{iHt}A_{l}e^{-iHt}$. Define $A_{l}:=A_{l}^{†}⊗A_{l}$ and similarly for B.

**Proposition** **1.**
*The isospectral twirling of the 4k−point OTOC is given by*
(9)$${\langle {OTOC}_{4k}\left(t\right)\rangle }_{G}=\mathrm{tr}\left({\tilde{T}}_{\pi }^{\left(4k\right)}\left({⊗}_{l=1}^{k}A_{l}{⊗}_{l=1}^{k}B_{l}\right){\hat{R}}^{\left(4k\right)}\left(U\right)\right)$$


See Appendix A for the proof. For k=1, we obtain the 4-point OTOC, see Section A.2:(10)$${\langle {\mathrm{OTOC}}_{4}\left(t\right)\rangle }_{G}=\mathrm{tr}\left({\tilde{T}}_{\left(1423\right)}\left(A⊗B\right){\hat{R}}^{\left(4\right)}\left(U\right)\right)$$

With the above formula, we compute the asymptotic value of the 4−point OTOC:(11)$$\begin{array}{ccc} \;{\langle {\mathrm{OTOC}}_{4}\rangle }_{Cl}\;\left(\infty \right)\; & = & \;\;\sum_{\pi \sigma }W_{g}^{+}\left(\pi \sigma \right)\mathrm{tr}\left(U_{\infty }^{⊗2,2}QT_{\sigma }\right)\\ \; & \times & \;\mathrm{tr}({\tilde{T}}_{\left(1423\right)}\left(A⊗B\right)QT_{\pi })\\ \; & + & \;W_{g}^{-}\left(\pi \sigma \right)\mathrm{tr}\left(U_{\infty }^{⊗2,2}Q^{⊥}T_{\sigma }\right)\\ \; & \times & \;\mathrm{tr}({\tilde{T}}_{\left(1423\right)}\left(A⊗B\right)Q^{⊥}T_{\pi }) \end{array}$$

**Proposition** **2.**
*By setting in (Equation 10) A and B as non-overlapping Pauli operators on qubits, the asymptotic value of the Cl−Isospectral twirling of the 4−point OTOC reads*
(12)$${\langle {OTOC}_{4}\rangle }_{Cl}\left(\infty \right)=\frac{2}{\left(d+2\right)}$$


See Appendix B for the proof. This value has to be compared with the asymptotic value for the isospectral twirling obtained by averaging the full unitary group:(13)$${\langle {\mathrm{OTOC}}_{4}\rangle }_{G}\left(\infty \right)=\frac{1}{\left(d+1)(d+3\right)}$$
showing a clear separation in the asymptotic decay of the OTOCs between the full Unitary and Clifford cases. For example, this shows that one cannot obtain the same asymptotic behavior by using only Clifford resources in a random quantum circuit.

Now we consider l−doped Hamiltonians by doping with T−gates, without loss of generality. Using the technique in [66], a lengthy but straightforward calculation gives the following theorem.

**Theorem** **1.**
*The asymptotic value of the averaged 4−point OTOC for an l−doped Hamiltonian reads:*
(14)$$\lim_{t\to \infty }{\bar{{\langle {OTOC}_{4}\left(t\right)\rangle }_{C_{l}}}}^{P(E_{i})}={\left(\frac{3}{4}\right)}^{l}\frac{2}{d}+\frac{1}{d^{2}}+\Omega \left(d^{-3}\right)$$
*As we can see, this result interpolates between the Clifford and Haar cases of Equations (Equation 12) and (Equation 13).*


The above equation shows a crossover between integrable and quantum chaotic behavior. The first term shows the integrable scaling of the OTOCs, while the second term shows the universal Haar chaotic behavior, see Proposition 2 above. The integrable term is exponentially suppressed in the doping parameter *l*. As a corollary, we obtain that iff l=Ω(n), l−doped stabilizer Hamiltonians attain the same scaling of *Haar* Hamiltonians for the infinite time 4−OTOCs.

## 5. Finite Time Behavior

From now on, we are concerned with finite-time behavior. Such behavior is ruled by spectral properties, while the eigenvectors are being chosen to be Haar-like, that is, we use the Haar-isospectral twirling (Equation 1). We will see that, insofar only the properties of the spectrum of the Hamiltonian *H* are concerned, different ensembles of spectra associated with different RMT distinguish the temporal profile of the chaos probes in the transient before the onset of the asymptotic behavior, which is the same for all the ensembles of spectra with a Schwartzian probability distribution [67]. By averaging over the unitary group in Equation (Equation 1), we have, on the one hand, effectively erased any information coming from the eigenstates of the Hamiltonian, and on the other hand, already introduced some of the properties of chaotic or ergodic Hamiltonians. For instance, these eigenvectors typically obey the eigenstate thermalization hypothesis [1,2,4,68]. In fact, it is striking that the spectra should have any effect at all once we use random eigenvectors. In order to examine the spectral properties revealed in the finite time behavior, we need to consider the spectral form factors.

Taking the trace of Equation (Equation 1), one obtains the 2k−point spectral form factors: $\mathrm{tr}\left({\hat{R}}^{\left(2k\right)}\left(U\right)\right)={\left|\mathrm{tr}\left(U\right)\right|}^{2k}={\left(d+Q\left(t\right)\right)}^{k}$, which follows easily from the cyclic property of the trace and the fact that ∫dG=1. The object $Q\left(t\right)=\sum_{i\neq j}\cos\left[\left(E_{i}-E_{j}\right)t\right]$ [63] is related to the quantum advantage of the performance of random quantum batteries. For k=1,2 these spectral form factors read ${\left|\mathrm{tr}\left(U\right)\right|}^{2}=\sum_{i,j}e^{i(E_{i}-E_{j})t}$ and ${\left|\mathrm{tr}\left(U\right)\right|}^{4}=\sum_{i,j,k,l}e^{i(E_{i}+E_{j}-E_{k}-E_{l})t}$. More generally, consider the coefficients ${\tilde{c}}_{\pi }^{\left(2k\right)}\left(U\right):=\mathrm{tr}\left({\tilde{T}}_{\pi }^{\left(2k\right)}U^{⊗k,k}\right)$. After the twirling, all the information about the spectrum of the Hamiltonian *H* is encoded in the ${\tilde{c}}_{\pi }^{\left(2k\right)}\left(U\right)$. We see that the 2k−point spectral form factors come from the identity permutation π=e, such that $T_{e}^{\left(2k\right)}={1\;l}^{⊗2k}$. For k=2 and the permutation $T_{\pi }^{\left(4\right)}=T_{\left(12)(3)(4\right)}^{\left(4\right)}\equiv T_{\left(12\right)}^{\left(4\right)}$, we instead obtain another spectral form factor, namely ${\tilde{c}}_{\left(12\right)}^{\left(4\right)}\left(U\right)=d^{-3}\mathrm{tr}\left(U^{2}\right)\mathrm{tr}{\left(U^{†}\right)}^{2}=d^{-3}\sum_{i,j,k}e^{i(2E_{i}-E_{j}-E_{k})t}$, which we will be needing later. Spectral form factors only depend on the spectrum of *U*. In particular, those we listed only depend on the gaps in the spectrum of *H*. For k=2, we set up this lighter notation for objects that we will be frequently using: ${\tilde{c}}_{e}^{\left(2\right)}\equiv {\tilde{c}}_{2}$, ${\tilde{c}}_{e}^{\left(4\right)}\equiv {\tilde{c}}_{4}$, ${\tilde{c}}_{\left(12\right)}^{\left(4\right)}\equiv {\tilde{c}}_{3}$. From now on, we will omit the order of permutations $T_{\pi }$. The operators ${\hat{R}}^{\left(2k\right)}\left(U\right)$ for k=1,2 are evaluated explicitly in [67].

In the following, we consider scalar functions P that depend on $U_{G}\equiv e^{-iG^{†}HGt}$. The isospectral twirling of P is given by ${\langle P(t)\rangle }_{G}=\int dG\;P\left(G^{†}UG\right)$. As we shall see, if $P_{O}$ is characterized by a bounded operator $O\in B(H^{⊗2k})$, we obtain expressions of the form ${\langle P_{O}\left(t\right)\rangle }_{G}=\mathrm{tr}\left[{\tilde{T}}_{\sigma }O{\hat{R}}^{\left(2k\right)}\left(t\right)\right]$, where ${\tilde{T}}_{\sigma }$ is a normalized permutation operator, $\sigma \in S_{2k}$.

The average ${\langle P_{O}\left(t\right)\rangle }_{G}$ only depends on the spectrum of the generating Hamiltonian *H*. One can then average the value of ${\langle P_{O}\left(t\right)\rangle }_{G}$ over the spectra of an ensemble of Hamiltonians *E*. We denote such average as ${\bar{{\langle P_{O}\left(t\right)\rangle }_{G}}}^{E}$. One can observe that the average over the unitary group and average over the ensemble of Hamiltonians return a quantity that depends on the same eigenvector statistics but different eigenvalue statistics. In this procedure, we are neglecting the possible connection between the eigenvector statistics and the eigenvalue statistics, as suggested in the following papers [69,70]; this has been to highlight the role of how the spectrum is enough to distinguish the finite-time case. In the following, we are going to look at the following ensembles of Hamiltonians E≡GUE, E≡GDE or E≡P, where the first is a class of chaotic Hamiltonians, and the others are classes of integrable Hamiltonians. It is important to remark that GUE is not the only a class of chaotic Hamiltonians; in random matrix theory, there are also two other classes of chaotic Hamiltonians: the Gaussian Orthogonal Ensemble (GOE) and the Gaussian Symplectic Ensemble (GSE) [50]. In the paper, we avoided considering these two classes of chaotic Hamiltonians since the behavior of the spectral coefficients ${\bar{{\tilde{c}}_{2}\left(t\right)}}^{E}$ and ${\bar{{\tilde{c}}_{4}\left(t\right)}}^{E}$ for GOE and GSE is not qualitatively different from the behavior of the coefficients for GUE [71]. Since the information about the spectrum of *H* is contained in the ${\tilde{c}}_{\pi }^{\left(2k\right)}\left(U\right)$, computing ${\bar{{\langle P_{O}\left(t\right)\rangle }_{G}}}^{E}$ requires the knowledge of ${\bar{{\tilde{c}}_{\pi }^{\left(2k\right)}\left(U\right)}}^{E}$. The details of the random matrix calculations necessary to compute these quantities can be found in [67]. We present here in Figure 1 the temporal evolution of ${\bar{{\tilde{c}}_{4}\left(t\right)}}^{E}$, which is the most important factor for our goals. The 4−point spectral form factor ${\tilde{c}}_{4}$ is able to distinguish the chaotic (GUE) and the integrable (GDE, Poisson) regime via the system-size scaling *d*. Both GUE and Poisson reach the first minimum ${\bar{{\tilde{c}}_{4}\left(t\right)}}^{E}=O\left(d^{-2}\right)$ in a time t=O(1), while GDE reaches the asymptotic value $\lim_{t\to \infty }{\bar{{\tilde{c}}_{4}\left(t\right)}}^{\mathrm{GDE}}=d^{-3}\left(2d-1\right)$ in a time $t=O(\sqrt{\log d})$. We observe that GUE and Poisson present a quite different temporal profile: dropping below the asymptotic value, GUE reaches the dip ${\bar{{\tilde{c}}_{4}\left(t\right)}}^{\mathrm{GUE}}=O\left(d^{-3}\right)$ in a time $t=O(d^{1/2})$ and then it rises to the asymptotic value $\lim_{t\to \infty }{\bar{{\tilde{c}}_{4}\left(t\right)}}^{\mathrm{GUE}}=d^{-3}\left(2d-1\right)$ in a time $O(d^{-1})$; on the other hand, Poisson never goes below $\lim_{t\to \infty }{\bar{{\tilde{c}}_{4}\left(t\right)}}^{P}=d^{-3}\left(2d-1\right)$ reaching it in a time $O(d^{1/2})$.

In [54], the authors defined the twirling of the operator $U^{⊗k,k}$, where $U\in E_{t}^{\mathrm{GUE}}:=\left\{e^{-iHt}\;|\;H\in \mathrm{GUE}\right\}$, i.e., $\Phi_{E_{t}^{\mathrm{GUE}}}\left(U^{⊗k,k}\right)=\int dH\;U^{⊗k,k}$ with dH the unitarily invariant measure over the GUE ensemble of Hamiltonians. From $dH=d(W^{†}HW)$, with W∈U(H), taking the Haar average over *W*, one easily obtains: $\Phi_{E_{t}^{\mathrm{GUE}}}\left(U^{⊗t,t}\right)=\int dH\;{\hat{R}}^{\left(2k\right)}\left(U\right)$, i.e., the ensemble average of the isospectral twirling of Equation (Equation 1) over the GUE ensemble. This approach presents some limits of applicability; unlike the isospectral twirling, it only works for a unitarily invariant distribution of Hamiltonians. In particular, it would not allow us to distinguish GUE from the integrable distributions.

We now apply the isospectral twirling to probe quantum chaos. Let us first study the finite time behavior of the OTOCs.

By setting in (Equation 10) A and B to be non-overlapping Pauli operators on qubits, one finds [67]:(15)$${\langle {\mathrm{OTOC}}_{4}\left(t\right)\rangle }_{G}={\tilde{c}}_{4}\left(t\right)-d^{-2}+O\left(d^{-4}\right)$$As this result shows, the 4−point OTOCs distinguish chaotic from integrable behavior through the timescales dictated by ${\tilde{c}}_{4}$, see Figure 1. In a previous work [54], it was found ${\langle {\mathrm{OTOC}}_{4}\left(t\right)\rangle }_{\mathrm{GUE}}≃{\tilde{c}}_{4}\left(t\right)$; we instead remark the importance of the offset $d^{-2}$. Indeed, in [67], it is shown that after a time $O(d^{1/3})$ the 4−point spectral form factor: ${\bar{{\tilde{c}}_{4}\left(t\right)}}^{\mathrm{GUE}}=O\left(d^{-2}\right)$, which makes the two terms in Equation (Equation 15) comparable.

## 6. Randomness of the Ensemble $E_{H}$

A chaotic Hamiltonian should generate a random unitary according to the Haar measure. To this end, we ask how random the ensemble $E_{H}$ generated by *H* is, i.e., how much the unitaries $G^{†}UG$ replicate the Haar distribution. We quantify randomness by computing the *k*-th frame potential of the ensemble $E_{H}$ [46,47], defined as
(16)$$F_{E_{H}}^{\left(k\right)}=\int dG_{1}dG_{2}{\left|\mathrm{tr}\left(G_{1}^{†}U^{†}G_{1}G_{2}^{†}UG_{2}\right)\right|}^{2k}.$$We have the following proposition:

**Proposition** **3.**
*The frame potential of $E_{H}$ is the square Schatten 2−norm of the isospectral twirling Equation (Equation 1):*
(17)$$F_{E_{H}}^{\left(k\right)}={∥{\hat{R}}^{\left(2k\right)}\left(U\right)∥}_{2}^{2}=\mathrm{tr}\left({\tilde{T}}_{1↔2}\left({\hat{R}}^{†(2k)}⊗{\hat{R}}^{\left(2k\right)}\right)\right)$$
*where ${\tilde{T}}_{1↔2}$ is the swap operator between the first 2k copies of H and the second 2k copies.*


See Section C.1 for the proof. The Haar value $F_{\mathrm{Haar}}^{\left(k\right)}=k!$ is a lower bound to this quantity [47], that is, $F_{\mathrm{Haar}}^{\left(k\right)}\leq F_{E_{H}}^{\left(k\right)}$ so a larger value of the frame potential means less randomness.

**Proposition** **4.**
*The frame potential of $E_{H}$ obeys the following lower bound:*
(18)$$F_{E_{H}}^{\left(k\right)}\geq d^{-2k}{\left|\mathrm{tr}(U)\right|}^{4k}$$


The above result is useful to see if an ensemble deviates from the Haar distribution, see Section C.2 for the proof. Taking the infinite time average $E_{T}\left(\cdot \right)=\lim_{T\to \infty }T^{-1}\int_{0}^{T}\left(\cdot \right)dt$ of the r.h.s., we can calculate a lower bound for the asymptotic value of the frame potential.

**Proposition** **5.**
*If the spectrum of H is generic:*
(19)$$E_{T}\left[d^{-2k}{\left|\mathrm{tr}(U)\right|}^{4k}\right]=\left(2k\right)!+O\left(d^{-1}\right)$$


As we can see, it is far from the Haar value k!. The request for the spectrum being generic is a stronger form of non-resonance; see Section C.3 for the definition of generic spectrum and Section C.4 for the proof. The infinite time average shows that the asymptotic value is the same for GUE and GDE. On the other hand, the frame potential $F_{E_{H}}^{\left(k\right)}$ is non-trivial in its time evolution. For k=1, we have [67]:(20)$$F_{E_{H}}^{\left(1\right)}=\frac{d^{2}}{\left(d^{2}-1\right)}\left(d^{2}{\tilde{c}}_{4}\left(t\right)-2{\tilde{c}}_{2}\left(t\right)+1\right)$$
where of course the coefficients ${\tilde{c}}_{k}\left(t\right)$ do depend on the spectrum of *H*. We can now take the ensemble average ${\bar{F_{E_{H}}^{\left(1\right)}}}^{E}$ of this quantity. The results are plotted in Figure 2. We can see that the behavior of the Poisson and GDE spectra is quite distinct from that of the GUE. Indeed, for the first two ensembles, the frame potential never goes below the asymptotic value $3+O(d^{-1})$, so it always stays away from the Haar value 1. On the other hand, the frame potential, corresponding to GUE, equals the Haar value 1 in the whole temporal interval $t\in [O\left(d^{1/3}\right),O\left(d\right)]$.

**Proposition** **6.**
*The ensemble average of the frame potential for E≡GDE satisfies:*
(21)$${\bar{F_{E_{H}}^{\left(k\right)}}}^{\mathrm{GDE}}\geq \left(2k\right)!+O\left(d^{-1}\right)$$
*showing that the GDE ensemble is always different from the Haar value.*


See Section C.5 for the details of the proof.

## 7. Loschmidt Echo and OTOC

The Loschmidt Echo (LE) is a quantity that captures the sensitivity of the dynamics to small perturbations. In [72,73], it was found that under suitable conditions, the OTOC and LE are quantitatively equivalent. Our aim in this section is to give another insight in that direction, showing that, using the isospectral twirling, the LE assumes the form of an OTOC-like quantity. The LE is defined as the fidelity between quantum states [74,75,76]; for an infinite temperature state, we have that the LE is $L\left(t\right)=d^{-2}{\left|\mathrm{tr}\left(e^{iHt}e^{-i(H+\delta H)t}\right)\right|}^{2}$.

**Proposition** **7.**
*Let A∈U(H) be a unitary operator, provided that Sp(H)=Sp(H+δH) the isospectral twirling of the LE is given by:*
(22)$$\begin{array}{c} {\langle L(t)\rangle }_{G}=\mathrm{tr}\left({\tilde{T}}_{\left(14)(23\right)}A^{⊗2}{\hat{R}}^{\left(4\right)}\left(U\right)\right) \end{array}$$
*where*
$A:=A^{†}⊗A$
*. See Section D.1 for the proof. If one sets*
A
*to be a Pauli operator on qubits, one obtains [67]:*
(23)$${\langle L(t)\rangle }_{G}={\tilde{c}}_{4}\left(t\right)+d^{-2}+O\left(d^{-4}\right)$$


In conclusion, we can say that both LE and OTOC are proportional to the 4−point spectral form factor in this setting. We can conclude that also the LE is a probe of scrambling; we thus find an agreement with the statement of [54]. Indeed, in proving Equation (Equation 22), we give an expression of the LE in terms of the 2−point auto-correlation function $|\mathrm{tr}\left(A^{†}\left(t\right)A\right){|}^{2}$; in [54], it was proved that the decay of the averaged 2−point autocorrelation function implies the decay of the TMI, i.e., implies scrambling [12,13].

## 8. Entanglement

We now move on to showing how the isospectral twirling also describes the evolution of entanglement under a random Hamiltonian with a given spectrum. Consider the unitary time evolution of a state $\psi \in B(H_{A}⊗H_{B})$ by $\psi \mapsto \psi_{t}\equiv U\psi U^{†}$. The entanglement of $\psi_{t}$ in the given bipartition is computed by the 2−Rényi entropy $S_{2}=-\log\mathrm{tr}\left(\psi_{A}{\left(t\right)}^{2}\right)$, where $\psi_{A}\left(t\right):={\mathrm{tr}}_{B}\psi_{t}$.

**Proposition** **8.**
*The isospectral twirling of the 2−Renyi entropy is lower bounded by:*
(24)$$\;{\langle S_{2}\rangle }_{G}\geq -\log\mathrm{tr}\left(T_{\left(13)(24\right)}{\hat{R}}^{\left(4\right)}\left(U\right)\psi^{⊗2}⊗T_{\left(A\right)}\right)$$
*where $T_{\left(A\right)}\equiv T_{A}⊗{1\;l}_{B}^{⊗2}$ and $T_{A}$ is the swap operator on $H_{A}$.*


See Section E.1 for the proof. If one sets $d_{A}=d_{B}=\sqrt{d}$, one obtains [67]:(25)$$\begin{array}{ccc} \;{\langle S_{2}\rangle }_{G}\; & \geq & \;-\log\left[2d^{-1/2}+{\tilde{c}}_{4}\left(t\right)\left(\mathrm{tr}\left(\psi_{A}^{2}\right)-2d^{-1/2}\right)\right]\;\\ & + & \;O(1/d) \end{array}$$As the temporal behavior of ${\langle S_{2}\rangle }_{G}$ is dictated by ${\tilde{c}}_{4}\left(t\right)$, one expects that entanglement dynamics can also distinguish between chaotic and non-chaotic behavior. The complete analysis of these dynamics is found in [67].

## 9. Tripartite Mutual Information

The TMI is defined as [12,13] $I_{3}\left(A:C:D\right):=I\left(A:C\right)+I\left(A:D\right)-I\left(A:CD\right)$ where A,B and C,D are fixed bipartitions of past and future time slices of the quantum system after a unitary evolution *U*; I(A:C) is the mutual information defined through the Von Neumann entropy. Here, we work with the TMI using the 2-Rényi entropy as measure of entropy and denote it by $I_{3_{\left(2\right)}}\left(U\right)=\log d+\log\mathrm{tr}\rho_{AC}^{2}+\log\mathrm{tr}\rho_{AD}^{2}$; see Section F.2. Here, $\rho_{AC(AD)}={\mathrm{tr}}_{BD(BC)}\left(\rho_{U}\right)$, where $\rho_{U}$ is the Choi state [77] of the unitary evolution U≡exp{−iHt} and *H* a random Hamiltonian with a given spectrum. Set A=C and B=D, then, by defining $T_{\left(C\right)}^{U}:=U^{⊗2}T_{\left(C\right)}U^{†⊗2}$, $I_{3_{\left(2\right)}}$ can be written as (see Section F.3):(26)$$I_{3_{\left(2\right)}}\;\;=\;\;-3\log d+\log\mathrm{tr}\left(T_{\left(C\right)}^{U}T_{\left(C\right)}\right)+\log\mathrm{tr}\left(T_{\left(C\right)}^{U}T_{\left(D\right)}\right)$$The second term of Equation (Equation 26) is similar to the entanglement of quantum evolutions defined in [78].

**Proposition** **9.**
*The isospectral twirling of $I_{3_{\left(2\right)}}$ is upper bounded by:*
(27)$$\begin{array}{ccc} \;{\langle I_{3_{\left(2\right)}}\rangle }_{G}\; & \leq & \;\log d+\log\mathrm{tr}({\tilde{T}}_{\left(13)(24\right)}{\hat{R}}^{\left(4\right)}\left(U\right)T_{\left(C\right)}^{⊗2})\;\\ \; & + & \;\log\mathrm{tr}({\tilde{T}}_{\left(13)(24\right)}{\hat{R}}^{\left(4\right)}\left(U\right)T_{\left(C\right)}⊗T_{\left(D\right)}) \end{array}$$


Since the TMI is a negative-definite quantity, the decay of the r.h.s. of Equation (Equation 27) implies scrambling, i.e., the l.h.s. drops closer to its minimum value. The tightness of this bound deserves further investigations. By explicitly computing Equation (Equation 27), one has [67]:(28)I3(2)(t)G≤log2(2−3c˜4(t)+2Rec˜3(t))+log2(c˜4(t)+(2−c˜4(t))d−1)+O(d−2)We can now compute ${\bar{{\langle I_{3_{\left(2\right)}}\rangle }_{G}}}^{E}$ over the spectra GUE, GDE and Poisson. We set $d_{C}=\dim\left(H_{C}\right)$ and $d_{D}=\dim H_{D}$ and $d_{C}=d_{D}=\sqrt{d}$. The time evolution of ${\bar{{\langle I_{3_{\left(2\right)}}\rangle }_{G}}}^{E}$ depends on the spectral form factors. We can see in Figure 3 how the chaotic and integrable behaviors are clearly different. The salient timescales of $I_{3_{\left(2\right)}}\left(t\right)$ depend on the timescales of ${\tilde{c}}_{4}\left(t\right)$ [67]. The plateau values of Equation (Equation 28) are, for large *d*:(29)$$\lim_{t\to \infty }{\bar{{\langle I_{3_{\left(2\right)}}\left(t\right)\rangle }_{G}}}^{E}=2-\log_{2}d+O\left(d^{-1}\right)$$

One thing to note is that the fluctuations of GUE and Poisson decay in time
(30)$$t_{\mathrm{fluct}}=\alpha +\beta \log d$$
where the parameters α,β for the different ensembles are GUE: α=−3.9,β=0.8 Poisson α=−16.3,β=3.2.

## 10. Conclusions and Outlook

Chaos is an important subject in quantum many-body physics, and the understanding of its appearance is of fundamental importance for a number of situations ranging from quantum information algorithms to black hole physics. In this paper, we unified the plethora of probes to quantum chaos in the notion of isospectral twirling. Since this quantity depends explicitly on the spectrum of the Hamiltonian, one can compare its behavior for different spectra characterizing chaotic and non-chaotic behavior, which we did by using random matrix theory. We demonstrate how different temporal features depend on the interplay between spectrum and eigenvectors of the Hamiltonian. Random eigenvectors obtained with Clifford resources result in markedly different asymptotic values of the OTOCs. Moreover, a doping of Clifford circuits with non-Clifford resources interpolates the long time scaling of the OTOCs between a class of integrable models and quantum chaos.

In perspective, there are several open questions. First and foremost, we want to extend the results of the crossover to more structural aspects of the dynamics with the goal of obtaining a quantum KAM theorem. Second, one could systematically study how different spectra behave together with different ensembles of eigenvectors, for instance, interpolating between Clifford and universal resources in a random quantum circuit [79], by doping a stabilizer Hamiltonian with non-Clifford resources such as the T−gates [66]. Another important aspect is that of the locality of the interactions. In this work, we did not take into account the locality of interactions. Locality might result in even more striking differences in the onset of quantum chaotic behavior. In this paper, we have treated the spectrum and the eigenvectors of the Hamiltonian separately, showing how they both contribute to quantum chaotic features. This is possible because in the spectral resolution, spectrum and eigenvectors are distinct. However, in realistic systems, we often find that *both spectra and eigenvectors* possess quantum chaotic features; this should depend on the fact that we deal with local Hamiltonians. Through the connections found with entanglement and quantum thermodynamics, one also hopes to exploit these findings to design more efficient quantum batteries. Finally, the notion of isospectral twirling could be generalized to non-unitary quantum channels and used to study chaotic behavior in open quantum systems.

## Figures and Tables

**Figure 1 entropy-23-01073-f001:**
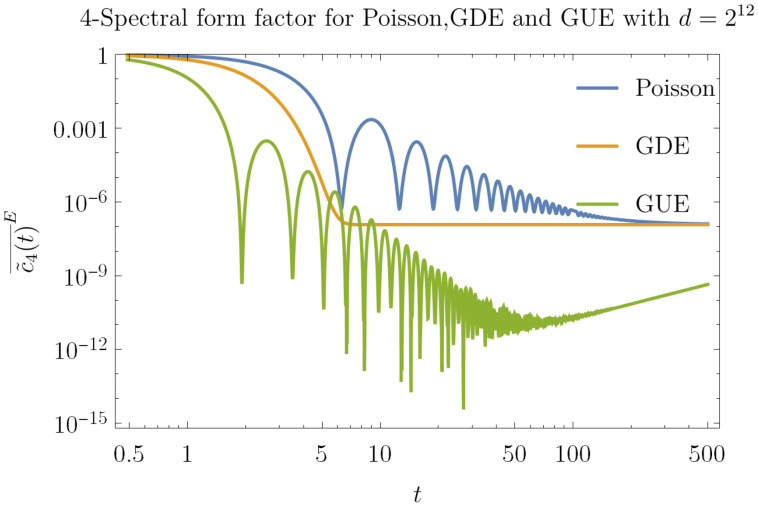
Log–log plot of the spectral form factor ${\bar{{\tilde{c}}_{4}\left(t\right)}}^{E}$ for different ensembles E ≡ P, E ≡GDE and E ≡GUE for $d=2^{12}$. The starting value is 1, while the asymptotic value is $\left(2d-1\right)d^{-3}$.

**Figure 2 entropy-23-01073-f002:**
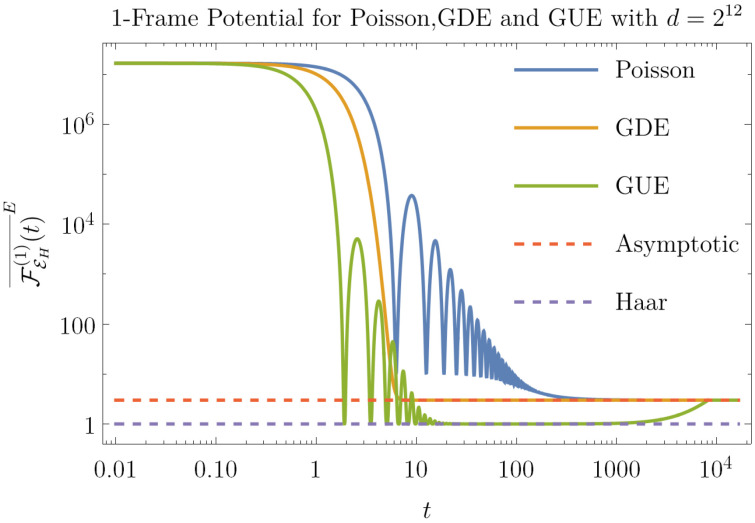
Log–log plot of the ensemble average of ${\bar{F_{E_{H}}^{\left(1\right)}\left(t\right)}}^{E}$ for E≡GUE, E≡GDE and E≡P for $d=2^{12}$. The dashed lines represent the Haar value $F_{\mathrm{Haar}}^{\left(1\right)}=1$ and the asymptotic value of $\lim_{t\to \infty }{\bar{F_{E_{H}}^{\left(1\right)}\left(t\right)}}^{E}=3+O\left(d^{-1}\right)$. Note that at late times t=O(d), ${\bar{F_{E_{H}}^{\left(1\right)}}}^{\mathrm{GUE}}$ distances from the Haar value [54] and reaches the asymptotic value.

**Figure 3 entropy-23-01073-f003:**
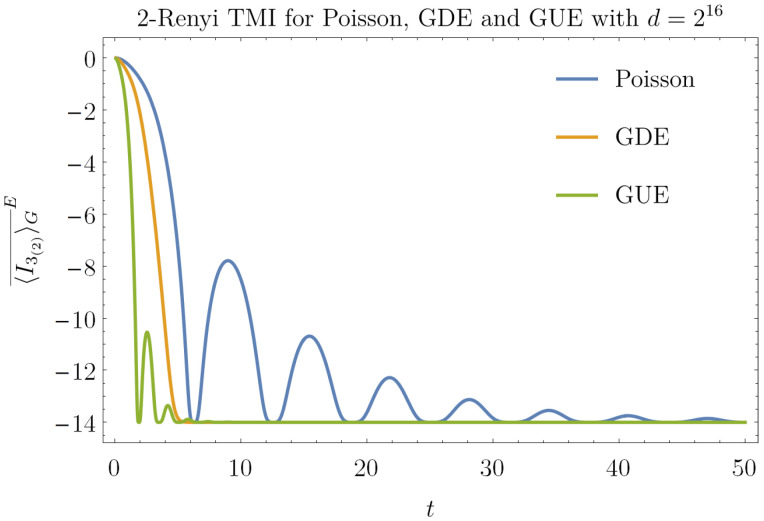
Plot of the upper bound for ${\bar{{\langle I_{3_{\left(2\right)}}\left(t\right)\rangle }_{G}}}^{E}$, see r.h.s of Equation (Equation 27), for E ≡ P, E ≡ GUE and E ≡ GDE with $d_{C}=d_{D}=d^{1/2}$ and $d=2^{16}$. GUE and Poisson reveal oscillations before the plateau whose amplitude and damp time increases with the system size *d*, see Equations (Equation 29) and (Equation 30), respectively. For GDE there are no oscillations: the plateau is reached in $O(\sqrt{\log d})$.

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
