# Peer review of "Isospectral Twirling and Quantum Chaos"

_entropy, 2021, doi:10.3390/e23081073_

Round 1
Reviewer 1 Report
The subject is important and I believe the authors have a contribution to it that deserves publication. They claim they have a new tool “the isospectral twirling” that unifies the other resources that are used to characterize quantum chaos. Although I am inclined to agree with them there is one thing that, I think, needs more elaboration in their manuscript. They identify the random matrix theory with the Gaussian Unitary Ensemble, that is the GUE. The GUE is just one of the three classes of random theory, and it is the one that applies in the case of systems without time invariance. For systems time invariant there are the Gaussian Orthogonal and the Gaussian Symplectic that apply, respectively, for the cases of systems with integer and half integer spins. I believe the authors are aware of this but they should explain why they only refer to the GUE.
I also would like to call the attention of the authors to an old question in random matrix theory, namely, if there is a connection between spectral and eigenvector statistics. By memory, I can cite two papers in which this problem has been addressed: C. M. Canali and V. E. Kravtsov, Phys. Rev. E 51 R5185(1995) and M. P. Pato, Phys. Rev. E 61 R3291(200).
Reviewer 2 Report
The article is devoted to problems related to the quantum chaos phenomena. The Authors concentrate on finding such formalism and an indicator of the quantum-chaotic effects that would be general enough to “include” in such formalism other witnesses of quantum chaos already known from the literature. They are frame potentials, Loschmidt echo – the fidelity, scrambling, and out-of-time-order correlators (OTOCs). In particular, the Authors propose a description with an application of a unified framework of the isospectral twirling. They propose to apply the Haar average of a k-fold unitary channel. The chaos’ witness presented in the manuscript depends on the spectrum of the Hamiltonian describing the system. It is a result of the fact that the operations in proposed formalism randomize over the eigenstates of a unitary channel U but leaves the spectrum invariant. The most relevant result presented in the article is that the computed isospectral twirling allows for the discrimination of the regular and chaotic characteristics of the system. In my opinion, the ideas and considerations presented in the manuscript are valid enough to be published.
The article is well written in general, and the considerations presented there seem to be correct. The Authors’ ideas and results are presented plainly, so the manuscript is accessible to a broad audience. Moreover, the Authors provide a comprehensive list of references to the works relevant in the field. Nevertheless, I suggest improving the paper on one point. Namely, the Authors mention the parameter referred as to Loschmidt echo. In fact, it is the fidelity between two wave functions commonly applied in quantum information theory. Moreover, fidelity was proposed as an indicator of quantum chaos a long time ago in Phys. Rev. A 30 (1984) 1610, Phys. Rev. Lett. 89 (2002) 214101, ibid 284102, and Phys. Lett. A 373 (2009) 1334. Practically, those papers were the first ones in which such a method was proposed. Thus, the Authors should mention this fact giving the references to the mentioned above articles.
Finally, I can state that the submitted manuscript deserves to be published and can be accepted after the minor amendment according to the above point.
